# Machine Learning Applied to Pre-Operative Computed-Tomography-Based Radiomic Features Can Accurately Differentiate Uterine Leiomyoma from Leiomyosarcoma: A Pilot Study

**DOI:** 10.3390/cancers16081570

**Published:** 2024-04-19

**Authors:** Miriam Santoro, Vladislav Zybin, Camelia Alexandra Coada, Giulia Mantovani, Giulia Paolani, Marco Di Stanislao, Cecilia Modolon, Stella Di Costanzo, Andrei Lebovici, Gloria Ravegnini, Antonio De Leo, Marco Tesei, Pietro Pasquini, Luigi Lovato, Alessio Giuseppe Morganti, Maria Abbondanza Pantaleo, Pierandrea De Iaco, Lidia Strigari, Anna Myriam Perrone

**Affiliations:** 1Department of Medical Physics, IRCCS Azienda Ospedaliero-Universitaria di Bologna, 40138 Bologna, Italy; miriam.santoro@aosp.bo.it (M.S.); giulia.paolani@aosp.bo.it (G.P.); lidia.strigari@aosp.bo.it (L.S.); 2Pediatric and Adult CardioThoracic and Vascular, Oncohematologic and Emergency Radiology Unit, IRCCS Azienda Ospedaliero-Universitaria di Bologna, 40138 Bologna, Italy; vladislav.zybin@studio.unibo.it (V.Z.); cecilia.modolon@aosp.bo.it (C.M.); luigi.lovato@aosp.bo.it (L.L.); 3University of Medicine and Pharmacy “Iuliu Hațieganu”, 400012 Cluj-Napoca, Romania; 4Division of Oncologic Gynecology, IRCCS Azienda Ospedaliero-Universitaria di Bologna, 40138 Bologna, Italy; giulia.mantovani18@unibo.it (G.M.); marco.distanislao@studio.unibo.it (M.D.S.); stella.dicostanzo@aosp.bo.it (S.D.C.); marco.tesei@aosp.bo.it (M.T.); pietro.pasquini@studio.unibo.it (P.P.);; 5Department of Medical and Surgical Sciences, University of Bologna, 40126 Bologna, Italy; antonio.deleo@unibo.it (A.D.L.); alessio.morganti2@unibo.it (A.G.M.); maria.pantaleo@unibo.it (M.A.P.); 6Radiology and Imaging Department, County Emergency Hospital, 400347 Cluj-Napoca, Romania; andrei1079@yahoo.com; 7Surgical Specialties Department, “Iuliu Hațieganu” University of Medicine and Pharmacy, 400012 Cluj-Napoca, Romania; 8Department of Pharmacy and Biotechnology, University of Bologna, 40126 Bologna, Italy; gloria.ravegnini2@unibo.it; 9Solid Tumor Molecular Pathology Laboratory, IRCCS Azienda Ospedaliero-Universitaria di Bologna, 40138 Bologna, Italy; 10Radiation Oncology, IRCCS Azienda Ospedaliero-Universitaria di Bologna, 40138 Bologna, Italy; 11Medical Oncology, IRCCS Azienda Ospedaliero-Universitaria di Bologna, 40138 Bologna, Italy

**Keywords:** artificial intelligence, radiomics, machine learning, computed tomography, sarcoma, leiomyoma, diagnosis

## Abstract

**Simple Summary:**

The differential diagnosis between uterine leiomyosarcomas and leiomyomas based on imaging represents one of the major challenges for gynecologists and radiologists. Currently, only histological examination can definitively resolve doubts in suspicious cases. The purpose of this study is to develop a machine learning model that can support clinical decision making. One of the proposed approaches, i.e., using the general linear model (GLM) classifier, has been patented by our team and has demonstrated good performance in retrospective analyses, with predicted area under the curve (AUC), sensitivity, and specificity on the test set ranging from 0.78 to 0.82, from 0.78 to 0.89, and from 0.67 to 0.87, respectively. The next step will involve validation at other medical centers and its prospective application.

**Abstract:**

Background: The accurate discrimination of uterine leiomyosarcomas and leiomyomas in a pre-operative setting remains a current challenge. To date, the diagnosis is made by a pathologist on the excised tumor. The aim of this study was to develop a machine learning algorithm using radiomic data extracted from contrast-enhanced computed tomography (CECT) images that could accurately distinguish leiomyosarcomas from leiomyomas. Methods: Pre-operative CECT images from patients submitted to surgery with a histological diagnosis of leiomyoma or leiomyosarcoma were used for the region of interest identification and radiomic feature extraction. Feature extraction was conducted using the PyRadiomics library, and three feature selection methods combined with the general linear model (GLM), random forest (RF), and support vector machine (SVM) classifiers were built, trained, and tested for the binary classification task (malignant vs. benign). In parallel, radiologists assessed the diagnosis with or without clinical data. Results: A total of 30 patients with leiomyosarcoma (mean age 59 years) and 35 patients with leiomyoma (mean age 48 years) were included in the study, comprising 30 and 51 lesions, respectively. Out of nine machine learning models, the three feature selection methods combined with the GLM and RF classifiers showed good performances, with predicted area under the curve (AUC), sensitivity, and specificity ranging from 0.78 to 0.97, from 0.78 to 1.00, and from 0.67 to 0.93, respectively, when compared to the results obtained from experienced radiologists when blinded to the clinical profile (AUC = 0.73 95%CI = 0.62–0.84), as well as when the clinical data were consulted (AUC = 0.75 95%CI = 0.65–0.85). Conclusions: CECT images integrated with radiomics have great potential in differentiating uterine leiomyomas from leiomyosarcomas. Such a tool can be used to mitigate the risks of eventual surgical spread in the case of leiomyosarcoma and allow for safer fertility-sparing treatment in patients with benign uterine lesions.

## 1. Introduction

Uterine sarcomas are rare, aggressive tumors that arise from the myometrium, accounting for 3–7% of uterine malignancies and approximately 1% of all female genital tract cancers; uterine leiomyosarcoma is the most common subtype of uterine sarcoma [1,2,3,4]. Treatment at an early stage typically requires hysterectomy by laparotomy to prevent neoplastic spread through the rupture and fragmentation of the tumor [5]. Uterine leiomyoma, on the other hand, is a commonly encountered and frequently asymptomatic benign condition that originates from the myometrium, with an incidence rate of 70–80% [6]. Surgery, when necessary, is typically performed laparoscopically if technically feasible, thus frequently with intraabdominal morcellation/fragmentation of the myoma [7,8].

Distinguishing between benign and malignant myometrial lesions is clinically important for planning an optimal management strategy (hysterectomy in leiomyosarcomas, fertility-sparing surgery, medical treatment, or no treatment in leiomyomas) and selecting the most appropriate surgical approach (laparotomy in leiomyosarcomas versus minimally invasive surgery in leiomyomas) [9,10,11]. This misdiagnosis can have a significant impact on the patient’s prognosis, as morcellated leiomyosarcomas can disseminate neoplastic cells in the abdomen, leading to iatrogenic abdominal sarcomatosis [12,13,14]. Not only high-power morcellation but also manual fragmentation of myometrial lesions should be avoided [15], particularly over the age of 50 years and always carefully weighing the risk–benefit ratio, as recently stated by the Food and Drug Administration (FDA) [16] and the American College of Obstetricians and Gynecologists (ACOG) [17]. Based on the 2017 Agency for Healthcare Research and Quality (AHRQ) report, which used the largest and most comprehensive dataset and rigorous analytic methods to determine estimates of prevalence of leiomyosarcoma, patients may be informed that the risk of unexpected leiomyosarcoma may range from 1 in 770 surgeries to less than 1 in 10,000 surgeries for presumed symptomatic leiomyomas [18].

Unfortunately, the diagnosis of leiomyosarcoma is currently made postoperatively through histological examination, since core biopsies, ultrasound scans, and radiological features cannot differentiate benign neoplastic lesions from malignant ones [19,20,21,22]. Transvaginal ultrasound is the first imaging modality employed to evaluate uterine smooth muscle tumors; although cheap and largely available, there is no ultrasound feature that has been univocally linked with malignancy, and the sensitivity for detecting leiomyosarcoma is low [19,20]. To date, for this purpose, magnetic resonance imaging (MRI) is the most reliable imaging modality, but it has not yet achieved optimal accuracy either and has high costs and limited availability [23].

On the other hand, computed tomography (CT) is commonly used for staging oncologic diseases as part of the pre-operative workup, including leiomyosarcomas [24], but has limited sensitivity and specificity in distinguishing benign myometrial lesions from malignant ones. A growing body of research has shown the role of radiomics in predicting patients’ prognosis for various types of tumors, including gynecological cancers [25,26,27,28]. However, regarding uterine sarcomas, our recent systematic review revealed limited evidence supporting the benefit of radiomics in this pathology for pre-operative evaluation, highlighting the need for further studies [29,30].

Therefore, in this pilot study, we sought to explore the potential role of contrast-enhanced CT features integrated with a radiomic analysis using machine learning models to differentiate leiomyosarcomas from leiomyomas.

## 2. Materials and Methods

### 2.1. Study Design and Study Population

This retrospective monocentric study was conducted at IRCCS Azienda Ospedaliero-Universitaria di Bologna, which is a European Reference on Rare Adult Solid Cancer (EURACAN) center. We included women with a histologic diagnosis of uterine leiomyosarcoma or leiomyoma submitted to surgery between 1 October 2011 and 30 September 2020, selected from the archives of the Division of Oncologic Gynecology. The inclusion criteria were as follows: (i) availability of abdominal CT in the Pictures Archiving and Communication System (PACS) records of our IRCCS; (ii) histologically confirmed uterine leiomyosarcoma or leiomyoma (excluding all forms of smooth uterine muscle of uncertain malignant potential (STUMP) or other types of sarcomas); (iii) CT scans obtained within two months before surgery for leiomyosarcomas and within six months for uterine leiomyomas. In our study, selected patients underwent CT evaluation before surgery in certain cases due to concerns regarding malignancy, which were based on clinical, ultrasound, and/or radiological features. These features included rapidly growing lesions, particularly in elderly patients, the absence of acoustic shadows, central vascularization, and a high color score. Histological examination confirmed malignancy in some cases where our suspicions were raised, while in others, the uterine lesion was found to be benign. For patients with no clear suspicion of malignancy, CT evaluation before surgery was conducted for other reasons, such as the differential diagnosis of abdominal symptoms, such as abdominal pain. Histological diagnosis was based on the 2020 WHO Classification of Female Genital Tumors [31]. An expert pathological review of all cases was performed by two dedicated gynecologic oncology pathologists. Exclusion criteria included previous neoplastic pathologies, pelvic radiotherapy, or chemotherapy. The study was approved by the local Ethics Committee 589/2022/OssAOUBo.

Data on age, body mass index, gravidity and parity, hormonal therapy usage, menopause, and clinical and treatment data were obtained by reviewing the electronic medical records. A database was created to include all clinical data of patients diagnosed with leiomyosarcomas, while basic medical history data were collected for patients with leiomyomas.

### 2.2. Contrast-Enhanced CT Imaging Details

Multiphasic CT images in DICOM format were collected from all patients. The imaging protocols included the following: (i) all CT examinations were internal; (ii) only the portal venous contrast-enhanced CT phase images of the abdomen and pelvis were considered eligible for the study.

The imaging parameters used were as follows: mA range: 37–105, kV range: 100–130, helical technique: helix; slice thickness range: 1.25–5 mm; and low-osmolality nonionic iodinated contrast agent administered IV dose: 90–140 mL. CT examinations were performed on the following machines: GE Lightspeed 16-slice, GE Lightspeed VCT (GE Healthcare, Waukesha, WI, USA), Siemens SOMATOM Sensation 64 Cardiac (Siemens Medical Solutions, Forchheim, Germany), Philips ingenuity CT 128, and Philips Brilliance 16-slice (Philips, Amsterdam, The Netherlands).

### 2.3. Image Analysis

Patients were stripped of their identifying information and assigned a random subject number. Three radiologists with experience in gynecological, pelvic, and abdominal imaging, blinded to the clinical and histological data, assessed the eligibility of the images. The CT scans were evaluated by the same dedicated radiologists and classified through the application of a score based on dimensions and features of the lesions that includes 9 items regarding CT lesion features and 1 item regarding clinical suspicion (Table 1). The items for the scoring system were selected by radiologists and gynecologists together, to identify the CT characteristics more likely associated with benign lesions or with malignant ones. The radiologists classified the lesions as probably benign or malignant. The first evaluation of the CT scans was conducted masking the clinical history and the final diagnosis of the corresponding cases. In the second step, a 30 min lecture based on epidemiological and clinical features of the two types of lesions (uterine leiomyoma and leiomyosarcoma) was given to the radiologists and a second, clinically unblinded re-evaluation of the CT scans was requested. In this phase, the clinical history of each patient was made available to the radiologists. In detail, the clinical file included all the information available at the time of the CT scan: a general evaluation of the patient’s medical history with detailed clinical signs and symptoms, a detailed objective exam, paraclinical investigations including routine bloodwork, and a transvaginal ultrasound. Thus, a second classification of the CT scans was performed using the same radiological score in the presence of the patient’s clinical data to identify whether clinical information helps in the differential diagnosis process. Throughout this phase, the radiologists were blinded to the histological diagnosis and to each other’s interpretations to simulate a real-life setting.

### 2.4. Radiomic Feature Extraction

Contrast-enhanced CT images were imported in the MIM software (v.7.1.4, MIM Software Inc. Cleveland, OH, USA). The radiologists and gynecologist semi-automatically drew the volumes of interest of the lesion(s) that showed contrast enhancement with the aim of including all viable tumor tissue. An ad hoc developed Python (v.3.8.3) [32] script including the PyRadiomics package [33] was used for feature extraction from the contoured lesions. PyRadiomics is an open-source library that offers the possibility to calculate 107 features subdivided into 8 different classes: First-Order Statistics, 3D Shape Based, 2D Shape Based, Gray Level Co-occurrence Matrix (GLCM), Gray Level Run Length Matrix (GLRLM), Gray Level Size Zone Matrix (GLSZM), Neighboring Gray Tone Difference Matrix (NGTDM), and Gray Level Dependence Matrix (GLDM). A pre-processing operation was performed with the Python script before the feature extraction since the CT images were acquired with different scanners and protocols. Specifically, the images were resampled to obtain an isotropic voxel spacing of 5 mm and density discretization using a fixed bin size of 25 HU. Moreover, the same features were extracted from derived images obtained after the application of several filters (i.e., wavelet, square, square root, logarithm, exponential, gradient) to the original image. In cases with multiple lesions present in the same uterus, each lesion was contoured and processed individually.

### 2.5. Machine Learning Classifier

An ad hoc script in R software (v.4.2) [34] was used to create three supervised machine-learning-based models, trained and tested for the binary classification task (malignant vs. benign, based on the histopathology results) with a 10-fold cross-validation approach to augment data and perform a more reliable analysis. Firstly, the features were scaled using the z-score; then, by generating random seeds to make the results repeatable, the database was divided with balanced output (i.e., histopathological results) into 70% training and 30% testing for 10 iterations. 

Three different methods were employed to perform the feature selection operation, thus implementing different models. Specifically, for each iteration in the training phase, we used the Boruta (i.e., a wrapper built around the random forest classification algorithm), recursive feature elimination (*rfe*), and Least Absolute Shrinkage and Selection Operator (LASSO) approaches, implemented in R with *Boruta* [35], *rfe* [36], and *glmnet* [37] functions, respectively. The optimal RFE model was obtained for each iteration after the optimization of hyperparameters, performed using the *rfeControl* function with a 10-fold cv method, while the optimal LASSO model having the best lambda hyperparameter was obtained after a 10-fold cv with the *cv.glmnet* function. For each model, the features obtained from each of the ten iterations were stored. Moreover, according to the methodology presented by Van Timmermen et al. [38], since *boruta* and *rfe* functions do not consider the collinearity and the correlation between variables, the features selected by these models were further reduced with correlation analysis. Clusters of highly correlated features were detected setting a Person’s r cutoff ≥0.60, as suggested by Baessler et al. [39], and low correlated features from each cluster were retained. 

For each model, the selected features were combined in a general linear model (*glm* function [40]), in a support vector machine, i.e., SVM (*svm* function) [41], or in a random forest, i.e., RF (*randomForest* function) [42] machine-learning-based classifier. This operation was repeated 100 times by dividing the dataset into 70% training and 30% testing with balanced output. The prediction was performed on both the training and test sets by extracting the receiver operating characteristic (ROC) curve with area under the curve (AUC), sensitivity, specificity, and confidence intervals (CIs). In the test dataset, we used the same threshold obtained for the training dataset. For each approach, models with a statistically significant AUC (i.e., with CIs between 0.5 and 1) in both the training and test sets were selected and the Bayesian information criterion (BIC) was used to identify the optimal model. Finally, the ROC-derived parameters were used to compare the GLM-based optimal classifiers with the SVM and RF ones. The comparison of radiomic features selected by the optimal GLM machine-learning-based classifiers was performed using Pearson’s correlation test. The analysis scheme is shown in Figure 1. 

### 2.6. Ablation Study

An ablation study was conducted to assess how each type of radiomic feature contributed to each machine-learning-based model. Radiomic features were clustered as shape, first order, and texture according to their class. Each cluster remaining after the feature selection operation was removed from the input variables (e.g., “Delete first order”) or used individually (e.g., “Only first order”) as the input of the model classifier. The performances of the obtained classifiers were compared among themselves and with the “All Variable in” classifier (i.e., the classifier based on all the input variables) in terms of ROC-derived parameters.

### 2.7. Statistical Analysis

Statistical analysis was performed using R version 4.2 [34]. The Shapiro–Wilk test was used to test for normality. Nominal variables were reported using frequencies, while quantitative variables were described by mean and standard deviation. Differences between nominal variables were assessed using the Chi-square test, while for quantitative variables, Student t-tests and ANOVA tests were used. The *p*-value for statistical significance was set at 0.05. For all radiologists, ROC curves were generated and compared across readers as well as against chance. The power of the resulting AUCs was computed using the power.roc.test function of the pROC R package [43]. Krippendorff’s alpha test was used to estimate the diagnostic reliability of the radiologists. 

## 3. Results 

### 3.1. Study Cohort Characteristics 

The selection and number of patients with leiomyosarcoma are shown in Appendix A. The leiomyosarcoma group comprised 30 patients (mean age: 59 years, range: 30–83 years), while the leiomyoma group comprised 35 patients (mean age: 48 years, range: 28–72 years). Patients with a diagnosis of leiomyosarcoma had single lesions (30), while eight patients with a diagnosis of leiomyoma had multiple lesions (51 lesions in total, range 1–4). Population characteristics of all patients included in this study are presented in Appendix A. Patients diagnosed with leiomyoma were younger than those diagnosed with leiomyosarcoma (*p* < 0.001). Pathological, oncological, and follow-up data of patients with leiomyosarcoma are reported in Appendix A. Most patients were in stage I. Median progression time was 10.12 months and median survival time was 60.87 (Appendix A). 

Pre-operative CT accuracy in discriminating leiomyosarcomas and leiomyomas is reported in Table 2. The false-positive rate was 0.29, corresponding to a final histological diagnosis of leiomyoma when leiomyosarcoma was suspected pre-operatively. The false-negative rate was 0.13, meaning a diagnosis of leiomyoma was made although the true diagnosis was leiomyosarcoma. We also assessed the surgical delay in the misdiagnosed cases having a histological diagnosis of leiomyosarcoma (false negatives). The mean delay of surgery was 17 days compared to the group of correctly diagnosed leiomyosarcomas, although these data did not reach statistical significance.

### 3.2. Radiologist Diagnosis Accuracy 

We sought to assess the diagnostic accuracy of three experienced radiologists based on the evaluation of CT scans alone and in the presence of the patient’s clinical data, as it would typically occur in a real-life setting. The AUCs obtained were 0.72 (95%CI 0.61–0.84), 0.78 (95%CI 0.68–0.88), and 0.70 (95%CI 0.58–0.81) when the diagnosis was made only on CT scans. After rendering the clinical data available for use during the CT scan evaluation, the AUC slightly increased in the case of two radiologists to 0.79 (95%CI 0.69–0.89) and 0.77 (95%CI 0.66–0.87), respectively (Figure 2A,B and Table 2). Using a scoring system (Table 1), the re-evaluation of CT imaging showed the same moderate discrimination capacity with AUCs of 0.64 (95%CI 0.51–0.78), 0.77 (95%CI 0.65–0.88), and 0.74 (95%CI 0.61–0.87) when scoring only the CT features and 0.69 (95%CI 0.56–0.82), 0.79 (95%CI 0.68–0.9), and 0.76 (95%CI 0.65–0.88) when the clinical suspect point was added (Figure 2C,D and Table 2). 

Neither the availability of clinical data nor the usage of the CT scoring system managed to increase the diagnostic accuracy (Table 2, Appendix A). Krippendorff’s alpha test showed a low agreement between the radiologists (alpha = 0.44 in the absence of clinical data; alpha = 0.41 in the presence of clinical data), further confirming the low reliability of diagnosis based on CT scans. 

### 3.3. Radiomic Analysis and Machine Learning Model 

All uterine mass images were semi-automatically contoured and later controlled and corrected, if required. The process of regions of interest drawing showed a quick learning curve: the mean time for contouring was 25 min/patient at the beginning of the CT series and 10 min/patient after the first half of the patients. 

A total of 1409 features were included in the model. Feature selection was performed sequentially using the three abovementioned feature selection approaches and the classifiers. The machine-learning-based model performances (specifying both the feature selection method and the classifier) for training and testing were extracted and are reported in Table 3. The radiomic features obtained from the optimal machine-learning-based classifiers are also reported in Table 3. The selected variables are produced by higher-order algorithms, revealing that the differential diagnostic signature (i.e., leiomyosarcoma) is associated with higher values for all the identified radiomic features except for the wavelet.HHH_glszm_ZonePercentage using the GLM approach.

When the feature selection approaches were combined with RF or SVM, a higher number of radiomic features was retained in the final models having AUC (95%CI) assessed in the test cohort up to 0.97 (0.90–1.00) and 0.80 (0.62–0.98), respectively. Thus, RF seemed to outperform the GLM and SVM approaches.

The ROC curves obtained for each machine-learning-based optimal classifier are shown in Figure 2E–G. In particular, Figure 2E reveals a similar performance for all the GLM-based models (AUCs between 0.78 and 0.82), Figure 2F shows a better and equal performance for all the SVM-based models (AUC = 0.97), and Figure 2G presents the worst performance for the RF-based models (AUCs between 0.69 and 0.8). The computed power of these results ranged between 0.78 and 0.99. Figure 3 shows the autocorrelation between the radiomic features selected by the investigated machine-learning-based models, confirming that the identified radiomic feature predictors from each model are correlated among them and suggesting that the differential diagnosis depends on the specific characteristics emerging from the contrast-enhanced CT images of this cohort of patients.

The results of the ablation study are shown in Appendix A. After the feature selection process, the LASSO approach identified only the texture features, while Boruta and RFE led to the selection of both first-order and texture features. Thus, the study ablation was conducted for the last two approaches. The complementarity among the two types of features improved the performance of the machine-learning-based model for the SVM and RF approaches, while using the texture group types alone produced a similar performance to that of the “all-in” approach for the GLM classifier model.

## 4. Discussion

In our pilot study, we successfully constructed and applied three machine-learning-based algorithms to radiomic data extracted from contrast-enhanced CT images. These algorithms showed promising discrimination between malignant and benign mesenchymal lesions of the uterus in a pre-operative setting, with a predictive accuracy of at least 0.7 for all the models except one with an AUC value of 0.69. The performance of these approaches was comparable (in two cases) or exceeded (in the remaining seven cases) the measured discriminatory capacity of experienced radiologists on the same images, both with and without the patient’s clinical data. Therefore, should these encouraging results be further validated in extended cohorts, these algorithms could potentially serve as a supporting tool for diagnosis in less experienced centers.

Currently, the differential diagnosis of leiomyosarcomas and uterine myomas relies on postsurgical histological specimens due to the limitations of performing biopsies. The high heterogeneity of the tumors and the risk of neoplastic dissemination during biopsy procedures restrict their use [44]. Despite attempts to identify radiological features that accurately discriminate between malignant and benign mesenchymal lesions of the uterus, the results have remained modest [45]. MRI is currently the most reliable imaging modality for characterizing uterine masses due to its superior contrast resolution for soft tissues [46]. Studies reported sensitivities of 0.59 for CT evaluation and 0.82 for MRI [24]. Unfortunately, there are no pathognomonic features to diagnose uterine leiomyosarcomas [23,24,46,47] and, although MRI can offer a better evaluation than CTs or ultrasonography, it cannot definitively exclude malignancy [48,49]. The AUCs obtained by our expert radiologists (0.7–0.78) exceeded those described in the literature for contrast-enhanced CT [24], likely attributed to their extensive experience and exposure to rare cases, as our center is specialized in such diseases.

When coupling artificial-intelligence-based algorithms with imaging techniques, we achieved higher AUC values. For example, Malek et al. obtained accuracy percentages of 96.2% and 100% using complex decision trees within a supervised machine learning framework [46]. Chiappa et al. obtained high diagnostic discrimination (AUC = 0.85) by applying machine learning to radiomic data extracted from ultrasonography images [50]. In our study, machine learning models based on contrast-enhanced CT identified a subset of candidate radiomic features with sensitivities ranging from 0.78 to 0.89. However, previous approaches, albeit highly accurate, required substantial computing resources and long analysis times. Additionally, a noteworthy concern is the persisting imbalance between the availability of MRI bookings and the demand, leading to treatment delays that could potentially hinder the prognosis for patients with leiomyosarcoma. Moreover, neither MRI nor ultrasonography can be used for disease extension evaluation [23,24]. In our study, by leveraging CT as a comprehensive tool for both differential diagnosis and disease extension evaluation, we can significantly simplify and optimize the oncological patient’s curative path. To the best of our knowledge, no other studies have shown such a high discrimination sensitivity in the differential diagnosis of leiomyosarcomas and leiomyomas using only contrast-enhanced CT imaging.

Moreover, we implemented nine machine-learning-based models obtained by the combination of three feature selection methods (i.e., Boruta, RFE, and LASSO), with three classifiers (i.e., GLM, RF, and SVM) having different characteristics/behavior according to their definitions. For example, GLM is the most well-known and “simple” machine learning algorithm, which is based on the linear combination of radiomic features, thus allowing an easy interpretability and explainability of the results since the weights of the radiomic features are directly proportional to the radiomic features’ importance in the model. In addition, RF relies on decision tree approaches, and SVM is based on finding a hyperplane in the N-dimensional space able to divide and classify the data. These algorithms, although being a powerful tool for solving machine learning problems, suffer from the “black box” effect [51]. Nevertheless, the three feature selection algorithms and three classifiers developed in this study were selected because of their efficiency, in line with the considerations made by Wang et al. [52].

Regarding the radiomic features extracted from CT images, our machine-learning-based models showed good discrimination in terms of AUC in the test cohort, supporting the advantage of the proposed approaches compared to the ones from expert clinicians. Of note, irrespective of the feature selection approaches, RF reached the highest AUC in the test cohort (95%CI), i.e., 0.97 (0.90, 1.00), followed by the GLM and SVM approaches. Wang et al. [52] reported similar results regarding the distinction between malignant and benign soft-tissue lesions using radiomic features extracted from MRI. Unfortunately, due to the limited number of patients in the test cohort, the AUC values of the proposed models were not statistically significantly different, and thus we cannot identify the optimal one.

Regarding the feature reduction process, the investigated approaches selected a relatively large proportion of texture features. In addition, our ablation study showed that the first-order radiomic features are of limited value in the differential diagnosis of leiomyosarcomas and uterine myomas as assessed by the AUC obtained by the machine learning models (i.e., RF and SVM) using these features alone. Overall, a possible approach for centers with less experience is the combination of the proposed feature selection models with the GLM to favor the interpretation and trustability of the tools from clinicians. Indeed, the GLM is less affected by overfitting or by the “black box” problems compared to the RF and SVM approaches [53,54].

In our data, the miss rate for leiomyosarcoma was 0.13 and accuracy did not exceed 0.78 in the context of a dichotomic diagnosis between leiomyosarcomas and leiomyomas by expert radiologists. Since 13% of patients with leiomyosarcoma were misdiagnosed, there was a mean delay of the surgery of 17 days with respect to the 30 days expected for the oncology path in our country. Of course, other symptoms, such as pain and bleeding, expedited the decision-making process, leading to timely hysterectomy within a two-month timeframe without any postponements. It is worth noting that in the case of asymptomatic patients, the delay could have been substantially longer, emphasizing the crucial role of a support tool for radiologists in accurately diagnosing leiomyosarcoma prior to surgery.

In a recent systematic review carried out by our group, we discussed the heterogeneity of the various studies using artificial intelligence models to differentially diagnose leiomyosarcoma from leiomyomas [29]. Some studies showed the superiority of artificial intelligence applied to MRI in diagnostic accuracy compared to expert radiologists. However, these studies analyzed different histologic types, including carcinosarcomas, as well as different uterine sarcoma types, such as leiomyosarcomas and sarcomas of the endometrial stroma, together. These tumors have different clinical behavior, prognosis, and treatment; therefore, this could have affected the findings [55]. To overcome this issue, we selected a homogenous cohort comprising patients with a histological diagnosis of leiomyosarcoma and leiomyoma.

With this study, we patented a diagnostic algorithm that can be used in the differential diagnosis between leiomyosarcomas and leiomyomas. One limitation of our study could be represented by the patients’ numerosity. Leiomyosarcoma is an uncommon form of cancer, so the reported numbers hold significant importance, especially considering our institute’s role as an oncological hub.

A possible consequence of this limitation is that we could not establish which of the machine learning models was superior. Nevertheless, the accuracy of the presented models could be further improved by re-training them on a larger cohort of patients. Additionally, for implementing the model in clinical practice as a computer-aided diagnostic system, it should be validated on an external cohort having a higher number of patients.

## 5. Conclusions

Our pilot study laid the foundation for a machine-learning-based diagnostic tool, facilitating standardized diagnoses and enabling the use of CT scans as a complementary decision-support resource in centers that may lack access to MRI or highly qualified radiologists.

## 6. Patents

The results from this work were presented to the Italian Patent and Trademark Office with the registration number 102023000013284.

## Figures and Tables

**Figure 1 cancers-16-01570-f001:**
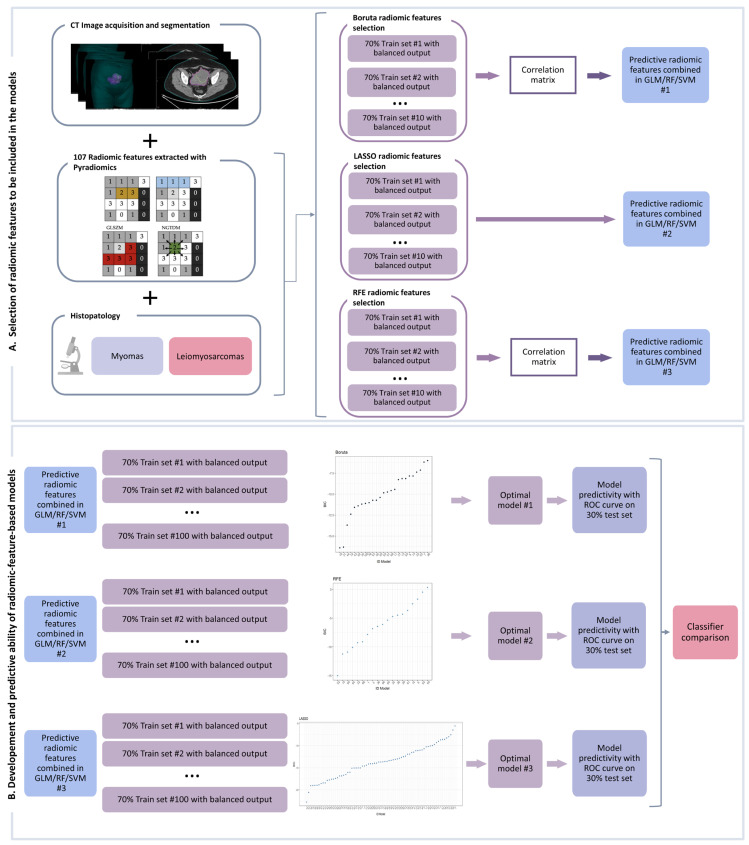
Workflow analysis composed of two steps: creation of the radiomic model (**A**) and comparison between the machine-learning-based optimal classifiers (**B**) as detailed in the manuscript. GLCM: Gray Level Co-occurrence Matrix; GLRM: Gray Level Run Length Matrix; GLSZM: Gray Level Size Zone Matrix; NGTDM: Neighboring Gray Tone Difference Matrix; GLM: generalized linear model; RF: random forest; SVM: support vector machine; ROC: receiver operating characteristic; LASSO: Least Absolute Shrinkage and Selection Operator; RFE: recursive feature elimination.

**Figure 2 cancers-16-01570-f002:**
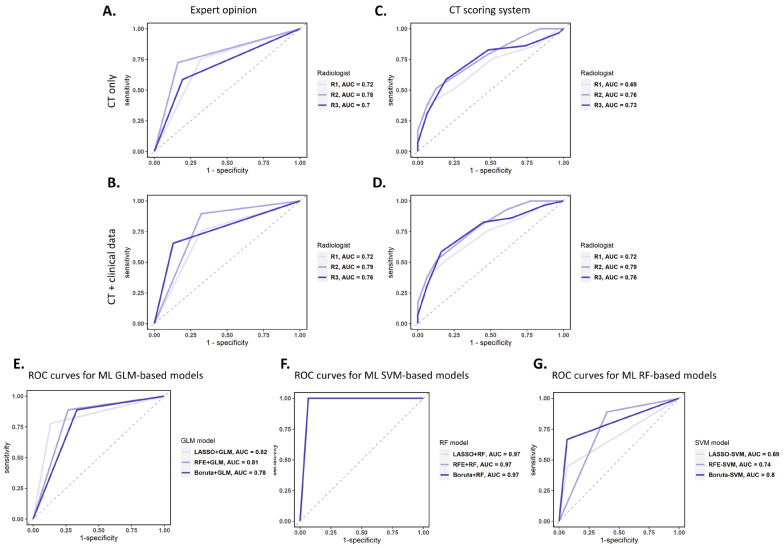
ROC curves for the diagnostic accuracy of radiologists’ expert opinions in the presence of CT scans only (**A**) and with clinical data at hand (**B**). ROC curves for the diagnostic accuracy of a CT scoring system containing nine CT items (**C**) and with clinical suspicion (**D**). (**E**–**G**) ROC curves on the test dataset for the diagnostic accuracy of the best GLM-based (**E**), SVM-based (**F**), and RF-based (**G**) classifier models for each radiomic feature selection method. AUC: area under the curve; ROC: receiver operating characteristic; CT: computed tomography; ML: machine learning; GLM: general linear model; SVM: support vector machine; RF: random forest; LASSO: Least Absolute Shrinkage and Selection Operator.

**Figure 3 cancers-16-01570-f003:**
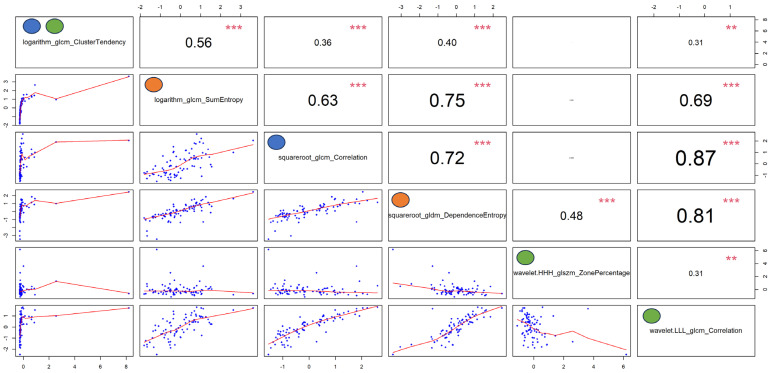
Autocorrelation plots of the features selected by the optimal models reported in Table 3. Pearson’s rho coefficients are reported in the upper right panels, while *p*-values (i.e., *p* < 0, 0.001, 0.01, 0.05, 0.1, and 1) are expressed as symbols (i.e., “***”, “**”, “*”, “.”, and “ “, respectively). The blue, orange, and green dots represent the features selected by the optimal Boruta, Least Absolute Shrinkage and Selection Operator (LASSO), and recursive feature elimination (RFE) machine-learning-based approaches, respectively.

**Table 1 cancers-16-01570-t001:** Radiological score includes 10 items based on dimensions and features of the lesions. We assigned 1 point for each feature with a greater malignant likelihood.

	Lesion Features	Points	
1	Maximal diameter: <10 cm/>10 cm	0	1
2	Number of lesions: Single/Multiple	1	0
3	Mass outline: Regular/Irregular	0	1
4	Margins: Well circumscribed/Ill defined	0	1
5	Hypodense basal or cystic areas: Absent/Present	0	1
6	Hyperdense basal areas: Absent/Present	0	1
7	Inhomogeneous contrast enhancement: Absent/Present	0	1
8	Adjacent organ infiltration: Absent/Present	0	1
9	Calcifications: Absent/Present	1	0
10	Clinical suspicion: Absent/Present	0	1
	Total score		

**Table 2 cancers-16-01570-t002:** Receiver operating characteristic (ROC) parameters of original presurgical diagnosis and summary of ROC analysis on diagnostic capacity of radiologists based on their expert opinions and a specific scoring system. Se: sensitivity; Sp: specificity; PPV: positive predictive value; NPV: negative predictive value; FPR: false-positive rate; FNR: false-negative rate; R1, R2, R3: radiologists’ pseudonyms; CT: computed tomography.

			Se	Sp	Accuracy	PPV	NPV	FPR	FNR
Presurgical diagnosis			0.87	0.71	0.78	0.72	0.86	0.29	0.13
Expert opinion	CT characteristics only	R1	0.72	0.84	0.78	0.81	0.76	0.16	0.28
		R2	0.59	0.81	0.70	0.74	0.68	0.19	0.41
		R3	0.76	0.68	0.72	0.69	0.75	0.32	0.24
		average	0.69	0.77	0.73	0.74	0.73	0.23	0.31
	CT characteristics and clinical profile	R1	0.90	0.68	0.78	0.72	0.88	0.32	0.10
		R2	0.66	0.87	0.77	0.83	0.73	0.13	0.34
		R3	0.76	0.68	0.72	0.69	0.75	0.32	0.24
		average	0.77	0.74	0.76	0.75	0.78	0.26	0.23
Diagnostic score	CT characteristics only	R1	0.52	0.87	0.70	0.79	0.66	0.13	0.48
		R2	0.59	0.81	0.70	0.74	0.68	0.19	0.41
		R3	0.38	0.94	0.67	0.85	0.62	0.06	0.62
		average	0.49	0.87	0.69	0.79	0.65	0.13	0.51
	CT and clinical susceptibility	R1	0.52	0.87	0.70	0.79	0.66	0.13	0.48
		R2	0.59	0.84	0.72	0.77	0.68	0.16	0.41
		R3	0.52	0.81	0.67	0.71	0.64	0.19	0.48
		average	0.54	0.84	0.69	0.76	0.66	0.16	0.46

**Table 3 cancers-16-01570-t003:** BIC and AUC values for training and testing of the statistically significant machine-learning-based models (i.e., 95%CI between 0.5 and 1) and radiomic features selected from the best models for each machine learning approach according to BIC value and their coefficient. LASSO: Least Absolute Shrinkage and Selection Operator; GLM: general linear model; RFE: recursive feature elimination; RF: random forest; SVM; support vector machine; AUC: area under the curve; Se: sensitivity; Sp: specificity; AIC: Aikake Information Criteria; BIC: Bayesian information criterion; coef: coefficient; NA: not available.

Model	AIC/BIC	AUC (95%CI) Train	Se/Sp Train	AUC (95%CI) Test	Se/Sp Test	Radiomic Feature	
						Name	Coef
LASSO + GLM	−36.39 −32.53	0.94 (0.88–0.99)	0.86 0.92	0.82 (0.65–0.99)	0.78 0.87	logarithm_glcm_SumEntropy	1.53
						squareroot_gldm_DependenceEntropy	2.15
Boruta + GLM	−32.32 −28.46	0.91 (0.82–0.99)	0.81 0.94	0.78 (0.62–0.94)	0.89 0.67	logarithm_glcm_ClusterTendency	19.24
						squareroot_glcm_Correlation	1.43
RFE + GLM	−30.70 −25.02	0.92 (0.84–1)	0.86 0.89	0.81 (0.65–0.97)	0.89 0.73	logarithm_glcm_ClusterTendency	13.17
						wavelet.HHH_glszm_ZonePercentage	−1.39
						wavelet.LLL_glcm_Correlation	1.37
LASSO + RF	NA	1.00 (1.00–1.00)	1.00 1.00	0.97 (0.90–1.00)	1.00 0.93	logarithm_glcm_SumEntropy, logarithm_glrlm_RunEntropy, squareroot_gldm_DependenceEntropy, wavelet.LLL_glcm_Correlation	NA
Boruta + RF	NA	1.00 (1.00–1.00)	1.00 1.00	0.97 (0.90–1.00)	1.00 0.93	logarithm_glcm_ClusterTendency, logarithm_glcm_MaximumProbability, squareroot_glcm_Correlation, wavelet.HHH_glszm_ZonePercentage, wavelet.HLL_firstorder_Energy	NA
RFE + RF	NA	1.00 (1.00–1.00)	1.00 1.00	0.97 (0.90–1.00)	1.00 0.93	logarithm_glcm_ClusterTendency, logarithm_glcm_MaximumProbability, logarithm_glszm_SmallAreaLowGrayLevelEmphasis, wavelet.HHH_glszm_ZonePercentage, wavelet.HLL_firstorder_Energy, wavelet.LLL_glcm_Correlation	NA
LASSO + SVM	NA	0.93 (0.87–1.00)	0.95 0.92	0.69 (0.50–0.87)	0.44 0.93	logarithm_glcm_SumEntropy, logarithm_glrlm_RunEntropy, squareroot_gldm_DependenceEntropy, wavelet.LLL_glcm_Correlation	NA
Boruta + SVM	NA	1.00 (1.00–1.00)	1.00 1.00	0.80 (0.62–0.98)	0.67 0.93	logarithm_glcm_ClusterTendency, logarithm_glcm_MaximumProbability, squareroot_glcm_Correlation, wavelet.HHH_glszm_ZonePercentage, wavelet.HLL_firstorder_Energy	NA
RFE + SVM	NA	1.00 (1.00–1.00)	1.00 1.00	0.74 (0.58–0.91)	0.89 0.60	logarithm_glcm_ClusterTendency, logarithm_glcm_MaximumProbability, logarithm_glszm_SmallAreaLowGrayLevelEmphasis, wavelet.HHH_glszm_ZonePercentage, wavelet.HLL_firstorder_Energy, wavelet.LLL_glcm_Correlation	NA

## Data Availability

The datasets used and/or analyzed during the current study are available from the corresponding author on reasonable request.

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
