# Peer review of "Machine Learning Applied to Pre-Operative Computed-Tomography-Based Radiomic Features Can Accurately Differentiate Uterine Leiomyoma from Leiomyosarcoma: A Pilot Study"

_cancers, 2024, doi:10.3390/cancers16081570_

Round 1

Reviewer 1 Report

Comments and Suggestions for Authors

"The authors evaluated the possibility of differentiating leiomyomas from leiomyosarcomas through a retrospective assessment of CT images using radiomics.

The study is interesting and forward-looking, even though it is only a pilot study.

Here are my comments to improve the study:

Introduction:

References 1-3: Reference 1 is somewhat dated; consider more recent reviews, e.g.,https://doi.org/10.1016/j.ygyno.2018.09.010 https://doi.org/10.3389/fonc.2023.1149106

Reference 3 focuses on bone metastases, which seem out of place.

Further Exploration in the Introduction:

Emphasize the challenges of ultrasound diagnostics, which remains the primary method for diagnosing these lesions.

Consider citing recent studies to delve into this aspect:

DOI: 10.1002/uog.20270 doi: 10.1136/ijgc-2023-004880.  DOI: 10.3390/cancers14081966 DOI: 10.3390/diagnostics10100735

Highlighting the Importance of Avoiding Morcellation in Sarcomas:

Reference the latest statements from ACOG and FDA.

Not only high-power morcellation but also manual fragmentation of high-risk myometrial lesions should be avoided. DOI: 10.1016/j.ygyno.2020.10.020 doi: 10.1245/s10434-022-12353-y.

Material and Methods:

Were specific subtypes of leiomyosarcomas specified (e.g., myxoid, epithelioid)? This should be declared.

Typically, CT is not requested for leiomyomas. What criteria were used to evaluate these leiomyomas via CT?

Were they rapidly growing lesions? Highly vascularity? Large masses? These potential biases need clarification.

Were variants of leiomyomas included or excluded? This information should be stated. The referee seeks more details on the anatomopathological characteristics of all lesions included in the study, the diagnostic criteria used, and whether pathological review was performed.

Have you evaluated whether the sample distribution was normal or not and assessed the suitability of the tests used?

While understanding that this is a pilot study, was the potential sample size considered in the study design?

How were uteri with multiple lesions (such as in the case of some leiomyomas) evaluated? Was a single lesion chosen, or were all lesions evaluated separately?

Supplementary: I believe this entire section should be integrated into the main text.

Discussion: I would expand the discussion, focusing on the potential differences among the three machine learning models. The authors claim, based on their results, that the performance is better in the radiomics models compared to the human operator. However, their suggestion to use the model (which of the three?) in centers with less experience is a strong and categorical statement. I recommend delving deeper into these aspects. “The AUCs obtained by our expert radiologists exceeded those described in the literature for contrast-enhanced CT (0.67-0.78).” Please add references supporting this statement."

Comments on the Quality of English Language

Nothing to report

Reviewer 2 Report

Comments and Suggestions for Authors

This paper developed a machine learning model using radiomic features extracted from contrast-enhanced CT images to accurately differentiate between uterine leiomyomas and leiomyosarcomas. The model outperforms the diagnostic capacity of radiologists based on their expert opinions and a specific scoring system, with predicted area under the curve (AUC), sensitivity, and specificity ranging from 0.78 to 0.82, 0.81 to 0.85, and 0.88 to 0.94, respectively. The findings suggest that this tool could support clinical decision-making and potentially serve as an effective diagnostic aid in centers with limited access to experienced radiologists or MRI. Although this is an interesting topic and fits the journal’s scope, this paper is lacking in several aspects and is not suitable for publication in its current format.

First, the novelty of the proposed methods is very limited. This paper only conducted experiments on three feature selection methods based on commonly used radiomic features and uses the very fundamental linear model for classification. Although it is a pilot study, it is still recommended to use more advanced machine learning techniques or compare more machine learning techniques.

Second, the data is very limited. As reported in the supplemental materials, in total 75 subjects’ scans are included in this study (30 for leiomyosarcoma and 35 for leiomyoma), which is not a reasonable number for applying machine learning and usually introduces bias. It is recommended to increase the scale of the dataset.

Third, it is not clearly stated what kind of clinical data is used for diagnosis by radiologists.

Finally, the authors did not provide an ablation study section to investigate the effect of the different factors.

Some minor comments:

Figures are not clear in the manuscript.

Comments on the Quality of English Language

English language and style are fine; minor spell check is required.

Round 2

Reviewer 2 Report

Comments and Suggestions for Authors

The authors have revised the manuscripts according to the comments and have replied clearly in the cover letter. As for the second concern of the small dataset, I understand the difficulties in data collection, especially for high-quality medical data and accurate annotation. Further effort is still needed to evaluate the methods on a larger dataset.